# EfficientNav: Towards On-Device Object-Goal Navigation with Navigation Map Caching and Retrieval

**Zebin Yang**[1,2]    **Sunjian Zheng**[3,4]    **Tong Xie**[1,2]    **Tianshi Xu**[1,2]    **Bo Yu**[3†]
**Fan Wang**[3]    **Jie Tang**[4]    **Shaoshan Liu**[3]    **Meng Li**[1,2,5†]

[1]Institute for Artificial Intelligence, Peking University
[2]School of Integrated Circuits, Peking University
[3]Shenzhen Institute of Artificial Intelligence and Robotics for Society
[4]School of Computer Science and Engineering, South China University of Technology
[5]Beijing Advanced Innovation Center for Integrated Circuits

## Abstract

Object-goal navigation (ObjNav) tasks an agent with navigating to the location of a specific object in an unseen environment. Embodied agents equipped with large language models (LLMs) and online constructed navigation maps can perform ObjNav in a zero-shot manner. However, existing agents heavily rely on giant LLMs on the cloud, e.g., GPT-4, while directly switching to small LLMs, e.g., LLaMA3.2-11b, suffer from significant success rate drops due to limited model capacity for understanding complex navigation maps, which prevents deploying ObjNav on local devices. At the same time, the long prompt introduced by the navigation map description will cause high planning latency on local devices. In this paper, we propose EfficientNav to enable on-device efficient LLM-based zero-shot ObjNav. To help the smaller LLMs better understand the environment, we propose semantics-aware memory retrieval to prune redundant information in navigation maps. To reduce planning latency, we propose discrete memory caching and attention-based memory clustering to efficiently save and re-use the KV cache. Extensive experimental results demonstrate that EfficientNav achieves 11.1% improvement in success rate on HM3D benchmark over GPT-4-based baselines, and demonstrates $6.7\times$ real-time latency reduction and $4.7\times$ end-to-end latency reduction over GPT-4 planner. Our code is available on https://github.com/PKU-SEC-Lab/EfficientNav.

## 1   Introduction

Aiming to navigate an agent to an object specified by its category in an unknown environment, object-goal navigation (ObjNav) presents a crucial yet challenging task for embodied agents [1, 58, 43, 25]. To enhance the generality of ObjNav, recent studies have proposed leveraging the common-sense reasoning capabilities of large language models (LLMs) to achieve long-term planning in a zero-shot manner [40, 4, 54, 23, 65]. Specifically, LLMs act as planners and determine a series of navigation sub-goals to guide the agent to the final destination [11]. In such systems, navigation goals, maps of explored areas, and current observations are typically provided as context for the LLMs [10, 91, 8]. This information can be conceived as memory for LLMs, providing information of explored areas and visited objects to facilitate planning and decision-making.

---

[†]Corresponding author. Emails: boyu@cuhk.edu.cn, meng.li@pku.edu.cn

39th Conference on Neural Information Processing Systems (NeurIPS 2025).

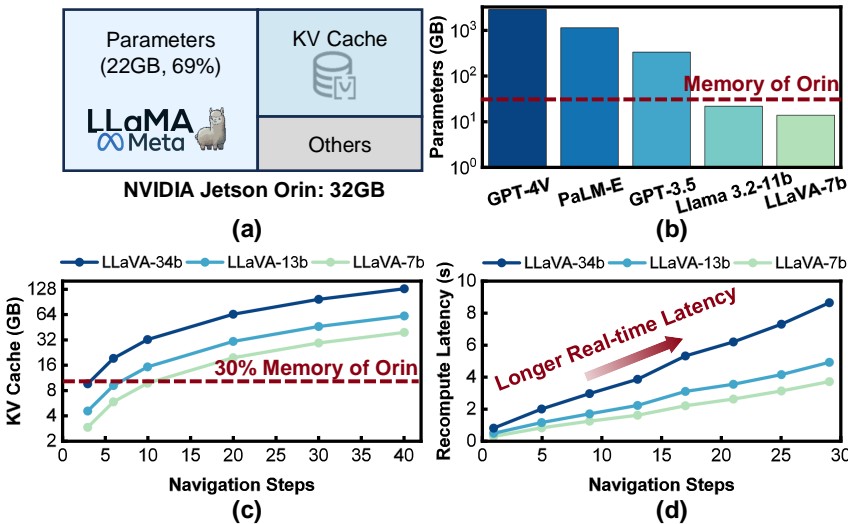

Figure 1: (a) Local devices (e.g., NVIDIA Jetson AGX Orin) exhibit constrained memory capacity (32GB). (b) Only smaller LLMs, such as Llama-3.2-11b, can be used as the planner due to memory limitations. (c) The KV cache of map information accumulates with navigation steps, exceeding the memory capacity. (d) Recomputing the KV cache incurs long real-time latency.

Though promising, LLM-based ObjNav systems face significant challenges due to the substantial memory requirements and computational complexity of LLMs. To ensure task success, giant LLMs, such as GPT-4, are typically employed, necessitating offloading these models to the cloud [63, 7, 91, 74, 66]. Such computing systems not only suffer from high communication latency, which degrades real-time performance [26, 53, 59, 44, 46, 36, 90], but also raise privacy concerns [18, 33, 52, 80, 16] and incur extremely high computational costs [5, 69]. Hence, a computing paradigm shift from offloading LLMs to the cloud towards accommodating LLMs on local servers or embedded devices, i.e., on-device computing, is required for ObjNav systems [24, 62].

However, enabling on-device ObjNav presents two primary design challenges, largely constrained by limited memory resources. First, only smaller LLMs can be deployed on local devices, thus causing planning performance drops. For example, as shown in Figure 1 (a) and (b), the NVIDIA Jetson AGX Orin [24] only has 32GB of DRAM, which can only accommodate smaller LLMs such as LLaMA-3.2-11b. Since larger LLMs typically have higher model capacity, enabling them to handle more complex and longer navigation tasks, on-device LLMs struggle to achieve the same level of performance as their cloud-based counterparts [8, 40]. Second, as context information accumulates during the navigation process, its corresponding KV-cache also grows and can eventually exceed the memory constraint (in Figure 1 (c), we set 30% of the Orin memory as the KV cache memory constraint, as the model weights also need to be stored in memory). And re-computing the KV-cache significantly slows down LLM decoding, introducing substantial compute latency (Figure 1 (d)).

In this paper, we propose EfficientNav, an on-device memory-augmented ObjNav system that enables efficient zero-shot in-door navigation. We observe that describing the navigation map can lead to long prompt lengths, and the corresponding KV-cache often cannot be fully stored due to memory constraints. Consequently, we select only a portion of the map description and load its KV cache for the LLM. However, the description selection will change the context order and the prefix of the corresponding KV cache. To enable reusing the retrieved KV cache despite the changing prefix, we propose discrete memory caching. This involves clustering the map information into groups and calculating the KV-cache for each group independently. Then, KV-caches within the same group are subject to cross-attention. To mitigate the impact of ignoring cross-attention between groups, we propose attention-based memory clustering. This technique uses LLM attention mechanisms to cluster related information into the same group. Furthermore, we observe that smaller LLMs may struggle to fully understand complex navigation maps. To address this, we propose semantics-aware memory retrieval to prune redundant map information during the group selection process, thereby improving the performance of smaller LLMs.

We summarize our contributions as follows: ❶ We propose discrete memory caching to prevent saving the KV cache of the whole map description and meet the memory constraints. ❷ We propose attention-based memory clustering to reduce the impact of ignoring cross-attention between groups. ❸ We propose semantics-aware memory retrieval to efficiently select the relevant groups and prune redundant map information. ❹ We conduct extensive experiments and show that EfficientNav can achieves 11.1% success rate improvements over GPT-4-based methods on HM3D dataset, and show 6.7× real-time latency and 4.7× end-to-end latency reduction over GPT-4 planner.

## 2 Related Work

**LLM-based object goal navigation.** According to the data processing pipeline, LLM-based ObjNav can be categorized into two paradigms: end-to-end and modular approaches. The end-to-end method, such as NaVid [83], directly converts RGB-D inputs into robot control policies, but incurs significant training costs [6, 27]. The modular approach, which is more prevalent due to simplicity, involves using LLMs as a high-level planner, and a lower-level controller handled by the robot itself. [40, 1, 54, 7, 23]. The LLM planner generates sub-goals to reach the final goal conditioned on the environment and instructions. The controller [56, 58], usually a small neural network, finds a trajectory from the current place to the sub-goal and controls the robot's motion in a real-time manner. We focus on optimizing the LLM planner, since it is the primary efficiency bottleneck due to its long inference latency and high memory cost [76, 10].

**Memory Augmentation for object goal navigation.** In zero-shot ObjNav tasks, the robot faces inefficiencies during sub-goal planning due to its limited field of view [7]. To achieve efficient environment exploration, mechanisms for memorizing navigation history are crucial for ObjNav [86, 87, 60, 88]. Maps built during navigation are typically leveraged to explicitly memorize the semantics and locations of visited areas and objects. While occupancy maps [40] can be used for memorization, graph-based navigation maps [10, 1, 91, 68] are increasingly

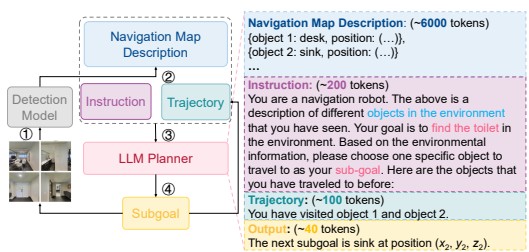

Figure 2: Software flow of LLM-based ObjNav.

favored for their ability to explicitly represent the semantics and locations of visited objects, which can be seamlessly integrated with LLM-based ObjNav [8, 47]. Hence, in this paper, we focus on the graph-based navigation maps.

**Efficient LLMs.** Many approaches have been proposed to enable efficient LLM inference on devices with limited computational resources, including quantization [12, 34, 71, 28, 32, 17], pruning [41, 21, 14, 41, 45], knowledge distillation [35, 50, 73], etc. However, these methods are not specifically optimized for ObjNav. In particular, they do not address the trade-offs between LLM size and the accuracy of the task planner, nor the challenge of efficient KV-cache saving and retrieval as the navigated map expands in ObjNav scenarios. We will further discuss them in Section 3.1.

**Comparisons with prior methods.** Our approach employs a modular architecture that uses a graph-based navigation map to represent the semantic and spatial information of the navigated area. It leverages LLMs to reason based on this graph, current observations, and the goal to generate the next navigation target. While aligned with many prior architectural designs, our approach is distinguished by its efficient and highly accurate on-device ObjNav, as shown in Table 1. This is achieved through our efficient KV-cache retrieval and pruning mechanism. Notably, our method is orthogonal to and compatible with classic efficient LLM techniques such as quantization, knowledge distillation, etc.

Table 1: Comparison with prior methods.

| Method | Zero-shot | LLM | On-device Inference | Memory Augmented |
|---|---|---|---|---|
| ViKiNG [55] | ✗ | - | ✓ | ✓ |
| NaVid [83] | ✗ | Vicuna | ✓ | ✗ |
| Skip-SCAR [39] | ✗ | - | ✓ | ✓ |
| Pixel Navigation [7] | ✓ | GPT-4 | ✗ | ✗ |
| InstructNav [40] | ✓ | GPT-4V | ✗ | ✓ |
| MapGPT [10] | ✓ | GPT-4 | ✗ | ✓ |
| LFG [54] | ✓ | GPT-4 | ✗ | ✓ |
| Imagine Before Go [88] | ✓ | GPT-4 | ✗ | ✓ |
| SayNav [49] | ✓ | GPT-4 | ✗ | ✓ |
| EfficientNav (**Ours**) | ✓ | LLaMA | ✓ | ✓ |

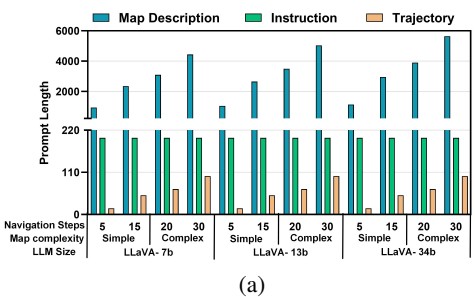
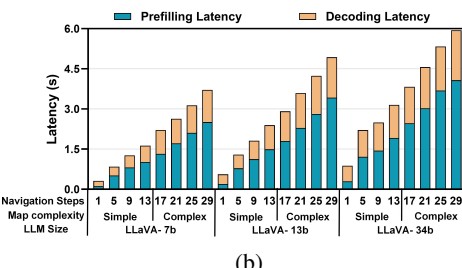

(a)                                                     (b)

Figure 3: (a) Average length of different parts of prompts across navigation steps. Prompt lengths vary slightly across models due to divergent sub-goal selections influencing map complexity. (b) On-device LLM planning latency across navigation steps.

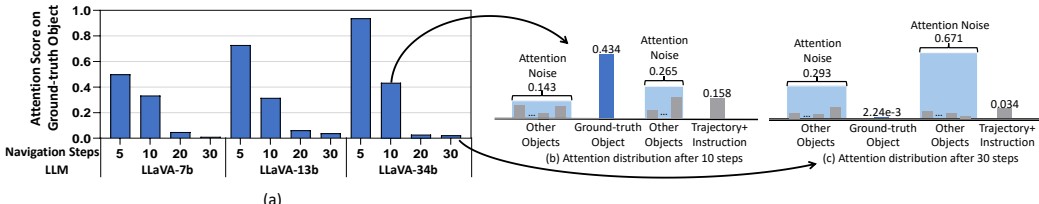

Figure 4: (a) Attention score on the most promising sub-goal (ground-truth object) in different navigation steps. Attention distribution after (b) 10 steps and (c) 30 steps using LLaVA-34b in sub-goal planning. As the amount of map information increases with the progression of navigation, the LLM can not fully understand the navigation map to focus on the most promising sub-goal. We observe similar problems in other small open-source LLMs such as LLaMA and Vicuna. The ground-truth object is chosen by GPT-4o.

## 3 EfficientNav: Memory Augmented On-device Navigation System

### 3.1 Challenges and Overview

**ObjNav software pipeline.** In our ObjNav, the goal-finding process is composed of iterative sub-goal finding tasks until the final goal is reached. Figure 2 illustrates our ObjNav's navigation pipeline for finding sub-goals and the ultimate goal. The first step involves generating semantic and distance information from RGB-D sensor captures using detection models, e.g., Grounding Dino [37]. Next, this semantic and spatial information is organized into the graph-based navigation map (we show an example in Appendix B). Third, the updated map and goal instructions are provided to the LLM for goal planning. Finally, if the target object is already present in the map, we choose it as the next goal; otherwise, the planner chooses a promising object in the map as the next sub-goal and continues detecting new environmental information after reaching the sub-goal. The computation challenges of this method lies in efficiently and accurately running the LLM as the map expands, which we discuss as follows.

**Challenge 1: tight memory constraints of local device limit the saving of KV cache of the navigation map description.** As discussed in Section 2 and shown in Figure 3 (a), with the progression of navigation, the amount of map information rapidly increases, thus introducing long prompt length, usually up to thousands of tokens. Re-computing the KV cache in each planning process will cause a long prefilling time, which is shown in Figure 3 (b). To avoid this cost, [22] saves the KV cache of history information in server memory, while as shown in Figure 1 (c), this can not meet the memory constraints of local devices. So a new memory caching mechanism is needed.

**Challenge 2: the tight memory constraint forces to use smaller LLMs, which have poorer model capacity and cause success rate degradation.** To meet the memory constraints of local devices, usually tens of gigabytes, we need to use smaller LLMs, e.g., LLaVA-7b. However, directly using them to replace giant LLMs, such as GPT-4, will cause a significant success rate drop [8, 29]. In practice, we find this comes from that the smaller LLM with lower capacity can not fully understand complex navigation maps. With the progression of navigation, the LLM can not pay attention to the

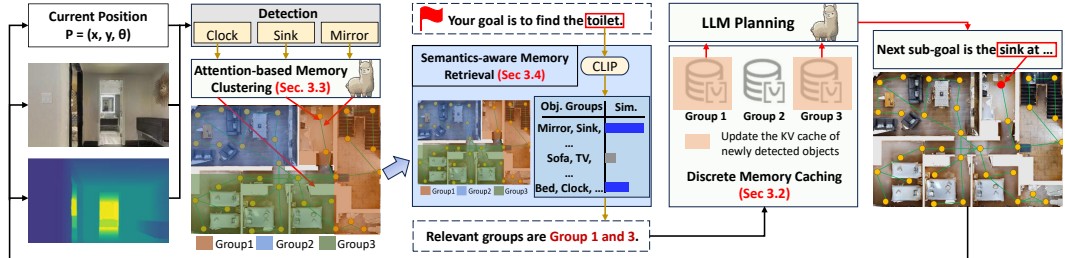

Figure 5: Overview of EfficientNav. Sim. stands for similarity.

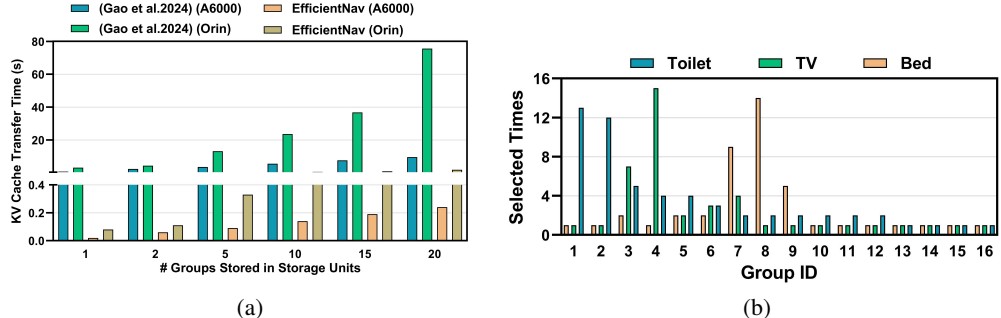

(a)                                                     (b)

Figure 6: (a) Memory transfer time of LLaVA-7b when using low-speed storage units. (b) For different final goals, the selected times of different groups are different, thus showing different relevance. For each final goal, the navigation task is conducted 15 times with different starting points. The groups are selected by GPT-4o.

important information among the huge amount of map information and choose the correct sub-goal, which is shown in Figure 4. So a memory selection strategy is needed to remove the redundant information for the smaller LLM.

**EfficientNav Overview.**   Based on these observations, we propose EfficientNav to enable efficient zero-shot ObjNav on local devices. Figure 5 demonstrates the overview of EfficientNav. To meet the memory constraints, we propose discrete memory caching to cluster objects into groups and calculate the KV cache of each group individually. Then we only select a portion of groups to LLM and load their KV cache to device memory. To accurately cluster information and improve success rate, we propose attention-based memory clustering, using LLM attention to guide group clustering. To help the smaller LLMs better understand the navigation map, in the group selection process, we propose semantic-aware memory retrieval to efficiently prune redundant map information.

## 3.2   Discrete Memory Caching

As discussed in Section 3.1, the memory constraints of local devices prevent full storage of the KV cache of navigation map descriptions. To mitigate this, [13] offloads the KV cache in high-capacity but low-speed storage units, e.g., CPU host memory for NVIDIA RTX A6000 or disk for Jetson Orin, and loads the KV cache on-demand during decoding. However, since local memory can not retain all the KV cache, this method requires repeated loading of the KV cache to device memory in each decoding phase. However, in each planning process, LLM needs to decode multiple tokens (around 40 in practice). The frequent communication between device memory and storage units will also cause huge latency, which is shown in Figure 6 (a).

To meet the memory constraint while avoiding re-computation and frequent memory transfers, we propose a discrete memory caching strategy. Building on [13], we store the KV cache in low-speed storage units and load the required KV cache into device memory, but introduce a critical optimization. As the navigation map contains redundancy, we only select partial important information according to the memory budget and load the KV cache only once, thus reducing memory transfer cost.

However, when selecting information, the order of context in the prompt will change. When pre-calculating the KV cache, existing works [38, 89] calculate the full attention of the whole context. In this approach, only the KV cache of its longest common prefix with the selected information can be

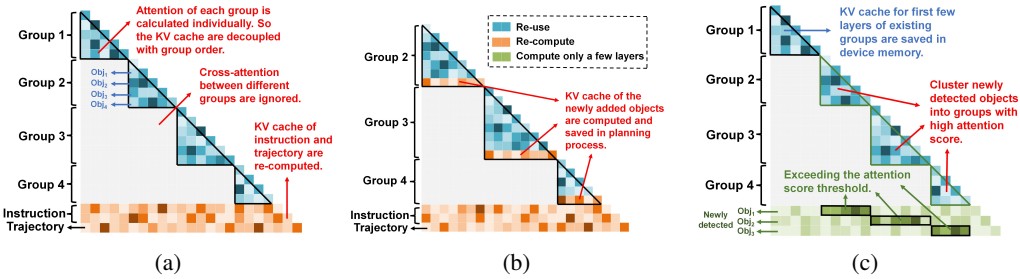

Figure 7: (a) When using discrete memory caching, the KV cache of existing groups can be reused in LLM planning. (b) When groups with newly added objects are selected during LLM planning process, the KV cache of these objects can be directly calculated and saved without impacting existing KV cache. (c) When using attention-based memory clustering, the newly detected objects will be clustered into different groups according to attention distribution.

reused. The KV cache after the changed position needs to be recomputed. To address this, we cluster objects in the navigation map into groups and compute KV cache of each group individually, as shown in Figure 7 (a). Then, only partial groups are selected to prompt the LLM planner. Hence, we can decouple the group KV cache and the group order, thus making full usage of the saved KV cache and avoiding re-computation. As shown in Figure 5, in each navigation step, newly detected objects will be added to existing groups, and we need to add their KV cache to the group's KV cache. To avoid changing context order within the group and impacting the existing KV cache, we concatenate the information of new objects at the end of the group information. As shown in Figure 7 (b), in the planning process, when groups with newly added objects are selected, the KV cache of these objects can be directly calculated and saved, without introducing extra computation. We follow the position encoding strategy in LLM inference as [15].

However, discrete memory caching will cause the ignorance of cross-attention between groups, leading to LLM performance drop [75, 19]. We also need to ensure that the groups with important information are not removed in group selection. Therefore, a correct memory clustering and memory selection mechanism is very important. We will explain our design in Section 3.3 and 3.4.

### 3.3 Attention-based Memory Clustering

As discussed in Section 3.2, to reduce the impact of ignoring cross-attention and maintain a high success rate, we need to cluster the information accurately and efficiently. Existing works [22, 75] partition context into uniform-length chunks, neglecting the relationships between objects in the navigation map. For example, the relationship between the oven and the pot is closer than the relationship between the oven and the bed. By grouping objects with closer relationships together and calculating their cross-attention, LLMs can better understand the navigation map and abstract the environment of a larger area. Meanwhile, the grouping granularity is also important. In contrast, if the granularity is too coarse, the KV cache size of each group will become large. Hence, only a few groups can be selected for the LLM, which will enhance the difficulty of selection and may neglect important information.

To adaptively control object clustering and group granularity, we use the attention of LLM itself to cluster the newly detected objects into different groups. As shown in Figure 7 (c), in the clustering process, we input the information of existing groups and the newly detected objects into LLM, and only infer a few layers (in practice, we find around $\frac{1}{10}$ layers of the whole model is enough). If the average attention between a newly detected object and an existing group exceeds a specific threshold, we cluster this object into the group. Otherwise, the remaining objects will be added to a new group. For the first group, we simply cluster the detected objects in the first navigation step into a group.

By using attention-based memory clustering, we can reduce the difference between discrete attention and full attention. Note that the clustering process will not introduce much extra computation and latency. This is because we only compute the first few layers of LLM, and the KV cache of existing groups has been pre-computed and its cache of the first few layers are kept in device memory.

## 3.4 Semantics-aware Memory Retrieval

As discussed in Section 3.1, the smaller LLM cannot fully understand complex navigation maps. Thus, a memory selection mechanism is needed to remove redundant information. [54] empirically selects environmental information based on object positions. However, we observe that when the final goal changes, the relevance of each group to the final goal also changes, as shown in Figure 6 (b). This empirical selection can not adapt to different final goals, thus showing lower performance. At the same time, it does not consider different device memory budgets. [8] uses LLM to select relevant information, while this will introduce extra LLM calls and long real-time latency.

Based on this, we propose semantics-aware memory retrieval, efficiently selecting groups according to semantic information to adapt to different sub-goals. As the task of removing redundant information is much easier than finding a specific sub-goal, to improve efficiency, we conduct group selection and sub-goal planning in a small-large model collaboration mode. For easier group selection, we use CLIP model [11] instead of LLM, which only has around 100M parameters and lower inference latency. For the harder sub-goal planning, we use LLM to select one specific object as the sub-goal. In the information selection process, we use CLIP to encode the object information in each group and the final goal. We then calculate the similarity between the encoding results of the final goal and each group, which is viewed as the probability for a group to include potential sub-goals. We consider the group relevant to the final goal only if the probability is larger than a threshold. To adapt to different devices with different memory budgets, we formulate the group selection as a knapsack problem:

$$\text{maximize} \quad P = \sum_{i=1}^{n} (P_i - threshold) \cdot x_i \tag{1}$$

$$\text{subject to} \quad \sum_{i=1}^{n} M_i \cdot x_i \leq M, \ \ x_i \in \{0,1\} \tag{2}$$

where $P_i$ denotes the probability to include sub-goals in group $i$, $M_i$ and $M$ denotes the memory usage of KV cache of group $i$ and device memory budget, $x_i$ denotes whether to select group $i$, $n$ denotes the number of groups. After solving the knapsack problem, we can select truly relevant information for LLM and meet the memory budget at the same time.

By using semantics-aware memory retrieval, we can efficiently select relevant groups adapting to different final goals and devices. Note that in each planning process, we only update the decoding results of the groups with newly detected objects. As the inference time of CLIP is much lower than LLM, the latency of the selection process is negligible. At the same time, between two adjacent planning processes, the navigation map will not change much. So it is likely that some group are both selected in the current step and last step. Hence, their KV cache has been loaded into device memory, thus reducing the KV cache loading time, which we will further show in our experiments.

## 4 Experiments

### 4.1 Experiment Setup

We evaluate EfficientNav on the HM3D dataset [3] based on the Habitat simulation platform [61]. In the simulation platform, the robot can access RGBD observation of the environment. In each task, the robot is placed at a different starting point in the environment and is only instructed to find a specific object, e.g., "TV", "chair", "sofa", "bed", "toilet", "plant" etc., which is harder than tasks giving detailed, step-by-step directions [2, 48]. For accuracy, we report two metrics: i) the average success rate (SR), and ii) the success rate penalized by path length (SPL), which both evaluates the accuracy and the efficiency of robot trajectory. For system efficiency, we report two metrics: i) real-time latency (RtL): the average robot planning time

Table 2: SR and SPL comparison.

| Method | Zero-shot | LLM | SR | SPL |
|---|---|---|---|---|
| DD-PPO [67] | ✗ | - | 27.9 | 14.2 |
| SemExp [9] | ✗ | - | 37.9 | 18.8 |
| Habitat-web [51] | ✗ | - | 41.5 | 16.0 |
| OVRL [72] | ✗ | - | 62.0 | 26.8 |
| ZSON [42] | ✓ | - | 25.5 | 12.6 |
| PixelNav [7] | ✓ | GPT-4 | 37.9 | 20.5 |
| ESC [92] | ✓ | - | 39.2 | 22.3 |
| VoroNav [68] | ✓ | GPT-3.5 | 42.0 | 26.0 |
| LLaVA Planner-34b [10] | ✓ | LLaVA-34b | 42.7 | 21.0 |
| L3MVN [79] | ✓ | RoBERTa-large | 50.4 | 23.1 |
| InstructNav [40] | ✓ | GPT-4V | 58.0 | 20.9 |
| LFG [54] | ✓ | GPT-4 | 68.9 | 36.0 |
| **EfficientNav-11b** | ✓ | LLaMA3.2-11b | **74.2** | **39.5** |
| **EfficientNav-34b** | ✓ | LLaVA-34b | **80.0** | **41.5** |

in one navigation step. ii) End-to-end latency (E2EL): the average latency for the robot to complete one navigation task (including multiple navigation steps and robot moving time). We implement our methods on 4 LLMs, LLaVA-7b, LLaVA-13b, LLaVA-34b, and LLaMA3.2-11b on NVIDIA A6000 GPU and Jetson Orin. We show the network architecture and implementation details in Appendix D and G. We also show an example of EfficientNav pipeline in Section B and C.

## 4.2  Main Results

**Comparison with state-of-the-art.** The comparison of SR and SPL is shown in Table 2. Compared with learning-based methods, EfficienNav achieves 18.0% SR and 14.7% SPL improvements over the prior-art method OVRL [72], without any training cost. This demonstrates the advantage of LLM-based navigation system, as discussed in Section 1. Compared with zero-shot methods, EfficientNav achieves 11.1% SR and 5.5% SPL improvements over LFG [54], which utilizes GPT-4. Compared with naive LLaVA-34b planner [10], EfficientNav achieves 37.3% SR and 20.5% SPL improvements. This is because we use semantics-aware memory retrieval, which helps the LLM focus on the most relevant groups and removes redundant information.

The latency comparison is shown in Table 3. Compared with GPT-4 planner [10], Efficient-Nav achieves 6.7× real-time latency reduction and 4.7× end-to-end latency reduction, by saving the communication time to the cloud server. Compared with naive LLaVA planner, EfficientNav achieves 8.8× and 6.5× real-time latency reduction and 3.7× and 4.4× end-to-end latency reduction on LLaMA3.2-11b and LLaVA-34b. This mainly comes from using discrete memory caching to avoid re-computation in the prefilling stage of planning. Prior-art so-

Table 3: Average latency comparison on A6000.

| Method | LLM | RtL | E2EL |
|---|---|---|---|
| GPT-4 Planner [10] | GPT-4 | 5.80s | 59.34s |
| LLaMA Planner-11b [10] | LLaMA3.2-11b | 3.07s | 46.40s |
| vllm [30] | LLaMA3.2-11b | 2.27s | 39.78s |
| EfficientNav-11b (**Ours**) | LLaMA3.2-11b | 0.35s | 12.70s |
| LLaVA Planner-34b [10] | LLaVA-34b | 5.63s | 55.32s |
| vllm [30] | LLaVA-34b | 4.43s | 47.95s |
| EfficientNav-34b (**Ours**) | LLaVA-34b | 0.87s | 12.51s |

lutions such as vllm can accelerate LLM inference, but they can not solve the problem of high re-computation cost, which is the main bottleneck. Compared with vllm, EfficientNav achieves 6.5× and 5.1× real-time latency reduction and 3.1× and 3.8× end-to-end latency reduction on LLaMA3.2-11b and LLaVA-34b.

**Real-time latency in different navigation steps.** The amount of map information increases in each navigation step. As shown in Figure 8, when using traditional serving methods [30], the increasing prompt length will introduce high recomputation cost and increasing real-time latency. However, when using EfficientNav, the real-time latency stabilizes after a certain navigation step, as we use semantics-aware memory retrieval to select partial groups for LLM according to memory budget. With the same amount of information given to LLM, EffcientNav also shows lower real-time latency, as discrete memory caching saves the re-computation time of LLM prefilling.

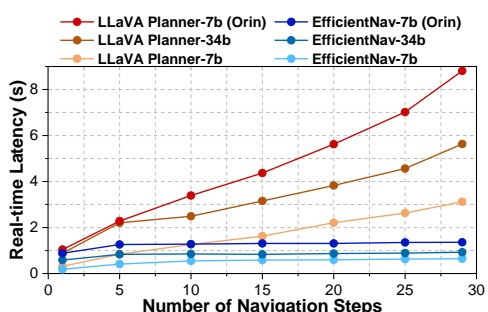

Figure 8: Comparison of real-time latency in different navigation steps on A6000 and Jetson Orin.

**Planning latency breakdown.** Figure 9 shows the latency breakdown of one navigation step. Compared to traditional serving methods [30], by avoiding re-computation, discrete memory caching can significantly reduce the prefilling time of planning by around 20×. And semantics-aware memory retrieval removes the redundant information, thus reducing the prompt length and reducing the computation cost and latency of the decoding stage of planning. The

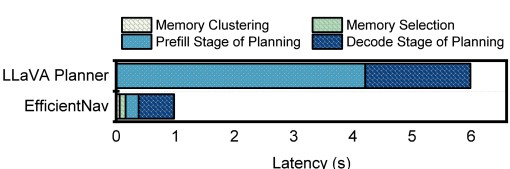

Figure 9: Planning latency (s) breakdown for LLaVA planner and EfficientNav on A6000.

latency of memory clustering and memory selection is negligible, as discussed in Section 3.3 and 3.4.

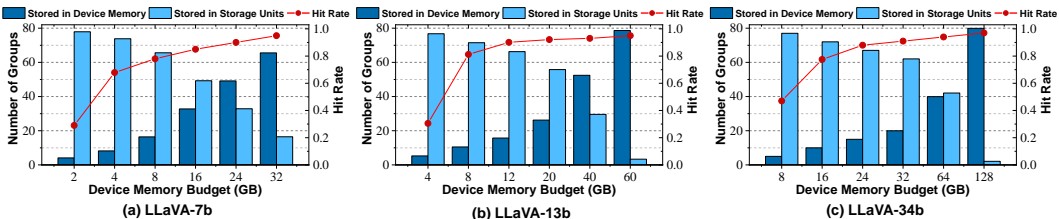

Figure 10: Comparison of memory caching distribution and memory cache hit rate across different device memory budgets for KV cache, using (a) LLaVA-7b, (b) LLaVA-13b, and (c) LLaVA-34b. Cache hit rate is defined as the proportion of selected groups that have been stored in device memory.

**Memory caching distribution and cache hit rate of EfficientNav.**  When using EfficientNav, in each planning process, we only select relevant groups for LLM. With larger device memory budgets for KV cache, we can store more groups in device memory. As discussed in Section 3.4, some selected groups may be just stored in device memory, so we do not need to reload these groups from storage units. We define the proportion of these groups as the cache hit rate. A high hit rate will lead to a low caching loading time. As shown in Figure 10, the cache hit rate of EfficientNav increases with the device memory budget and rapidly reaches a high level. This is because when the memory budget increases, more relevant groups are stored in device memory. At the same time, the semantics of each group will not change much between adjacent planning processes. So it is likely that some relevant groups are just loaded to device memory in the last few steps, leading to a high cache hit rate.

### 4.3 Ablation Study

**Individual influence of our methods.**  Table 4 shows the individual influence of our proposed methods. In naive LLaVA planner, we empirically cluster and select environmental information according to object positions [85, 54]. By using discrete memory caching, compared with naive LLaVA planner [10], we achieve 2.3× real-time latency and 1.5× end-to-end latency reduction, by re-using KV cache of navigation map description and avoiding re-computation.

Table 4: Ablation study on EfficientNav methods with LLaVA-34b.

| Method | SR | SPL | RtL | E2EL |
|---|---|---|---|---|
| LLaVA Planner | 42.7 | 21.0 | 5.63s | 55.32s |
| +Discrete Memory Caching | 43.1 | 21.0 | 2.42s | 36.94s |
| +Attention-based Memory Clustering | 63.3 | 34.2 | 2.32s | 32.58s |
| +Semantics-aware Memory Retrieval | 80.0 | 41.5 | 0.87s | 12.51s |

After attention-based memory clustering is used, objects in each selected group have high correlation, which helps the LLM to better understand the environment and reduces the impact of ignoring cross-attention at the same time, thus improving the success rate. When using semantics-aware memory retrieval, we can select the most related groups, which further improves the success rate. We also achieve 2.7× real-time latency and 2.6× end-to-end latency reduction, by selecting groups according to memory budget and avoiding frequent communication with storage units.

**Impact of device memory budget for KV cache.**  In semantics-aware memory retrieval, we select relevant groups according to the device memory budget. Here we evaluate the impact of memory budget on success rate and latency using LLaVA-34b. As shown in Table 5, on most cases, our method shows good robustness. However, when the memory budget for KV cache is too small, fewer relevant groups can be selected. Hence, the potential sub-goal may be ignored, which causes success rate drops and longer end-

Table 5: Ablation study on different device memory budgets for KV cache with LLaVA-34b.

| Memory Budget | SR | SPL | RtL | E2EL |
|---|---|---|---|---|
| 16GB | 74.7 | 37.3 | 0.59s | 14.24s |
| 24GB | 79.0 | 39.1 | 0.71s | 14.29s |
| 32GB | 80.0 | 41.5 | 0.87s | 12.51s |
| 40GB | 80.3 | 41.9 | 0.93s | 11.72s |

to-end latency. For a larger memory budget, more relevant groups can be selected. While the longer prompt length may cause longer planning time, which will slightly increase the real-time latency.

**Comparison with sparse attention approaches.**  Our discrete memory caching is similar to adaptive sparse attention methods [20, 31, 84]. However, the goals of general adaptive sparse attention and EfficientNav are different: EfficientNav clusters objects into different groups and calculates

the attention of each group individually. This structured sparse approach can decouple the group order and KV cache computation of each group, thus enabling KV reuse when we retrieve different groups in navigation planning and reducing real-time latency. However, for general adaptive sparse attention methods, their purpose is to minimize the difference between sparse

Table 6: Comparison with sparse attention approaches with LLaVA-34b.

| Method | SR | SPL | Real-time latency |
|---|---|---|---|
| Minference [20] | 35.3 | 18.0 | 3.88s |
| FlexPrefill [31] | 36.7 | 20.1 | 3.02s |
| EfficientNav | 80.0 | 41.5 | 0.87s |

attention and full attention and accelerate the attention calculation. The comparison with sparse attention approaches is shown in Table 6. Just using adaptive sparse attention methods shows huge accuracy drops. This is because these methods minimize the difference between sparse attention and full attention. But without memory retrieval, even full attention shows a low success rate, as discussed in Section 3.1. At the same time, without discrete memory caching, these methods need to recompute the KV cache of the navigation map description because of map changes, thus showing longer real-time latency.

## 5 Conclusion and Limitations

This work proposes EfficientNav, which enables efficient zero-shot ObjNav on local devices. We propose discrete memory caching to enable re-using KV cache and avoid re-computation in LLM planning. We also propose attention-based memory clustering to reduce the impact of ignoring cross-attention. To improve planning performance of smaller LLMs, we propose semantics-aware memory retrieval to remove the redundancy in navigation map. EfficientNav overcomes all existing baselines and enables running zero-shot ObjNav on local devices. While our study shows promising results, it has some limitations. We mainly focus on LLM-based navigation in this paper, while the inference speed of LLM can not reach that of small models even after acceleration. And EfficientNav should be carefully used in applications that need extremely low real-time latency.

## 6 Acknowledgments

This work was supported in part by NSFC under Grant 62495102, Grant 92464104, and Grant 62341407, in part by the National Key Research and Development Program under Grant 2024YFB4505004, in part by Beijing Municipal Science and Technology Program under Grant Z241100004224015, in part by 111 Project under Grant B18001, and in part by Longgang District Shenzhen's "Ten Action Plan" for Supporting Innovation Projects under Grant LGKCSDPT2024003 and LGKCSDPT2024004.

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

# A    LLM inference and KV Cache

The generative inference process of LLMs is divided into two stages, the prefill stage and the decode stage. In the prefill stage, the prompt sequence is taken as input $\boldsymbol{X}$, and GEMMs are performed to compute the query ($\boldsymbol{Q} = \boldsymbol{X}\boldsymbol{W}_Q$), key ($\boldsymbol{K} = \boldsymbol{X}\boldsymbol{W}_K$), and value ($\boldsymbol{V} = \boldsymbol{X}\boldsymbol{W}_V$) matrices for each Transformer block. These computed key and value matrices are cached as KV cache for subsequent steps. The attention output is derived as $\text{Attn}(\boldsymbol{Q}, \boldsymbol{K}, \boldsymbol{V}) = \text{softmax}(\frac{\boldsymbol{Q}\boldsymbol{K}^{\text{T}}}{\sqrt{D}})$, which further produces the first output token.

The decode stage iteratively generates one token per step via General Matrix-Vector operations. At step $t$, the newly generated token updates the cached Key and Value tensors through concatenation: $\boldsymbol{K}_t = \text{concat}(\boldsymbol{K}_{t-1}, \boldsymbol{K}_{\text{new}})$ and $\boldsymbol{V}_t = \text{concat}(\boldsymbol{V}_{t-1}, \boldsymbol{V}_{\text{new}})$. The updated cache enables efficient computation of self-attention over the extended sequence, yielding the next token. This autoregressive process continues until termination. Both stages leverage cached key-value pairs to avoid redundant computation.

# B    Object Navigation Flow

As discussed in Section 3.1, in our ObjNav, the goal-finding process is composed of iterative sub-goal finding tasks until the final goal is reached. Figure 11 shows an example of our ObjNAv. After reaching a sub-goal, the system generates semantic and distance information of the main objects from RGB-D sensor captures using detection models. Next, this semantic and spatial information is organized into the graph-based navigation map, and the updated map and goal instructions are provided to the LLM for goal planning. Here we also show an example of the LLM prompt and output in Section C.

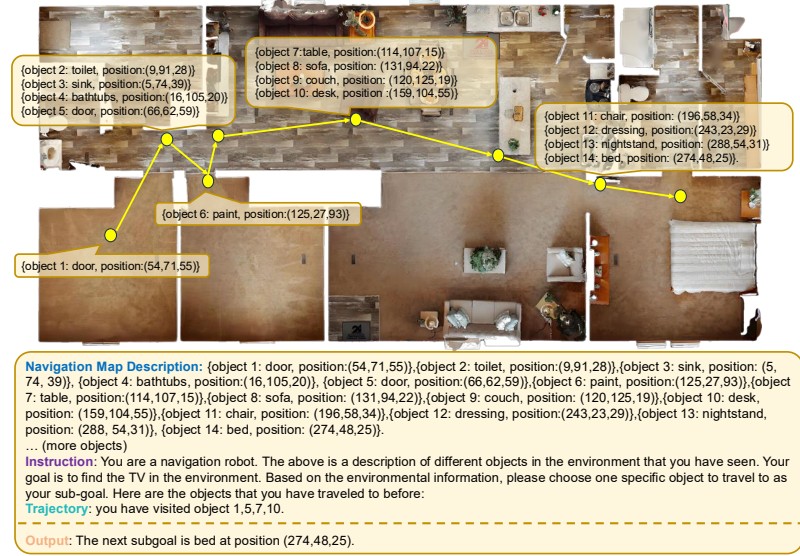

Figure 11: An example of our ObjNAv flow.

# C    Detailed Planning Example

We now further provide a more detailed example to illustrate the memory and prompt for EfficientNav.

**1. Navigation Map Description**: the navigation map records the objects and their position. To facilitate memory retrieval, we organize these objects into groups. Hence, an example of the navigation map is as follows:

> **Navigation Map Description**
>
> **Object Group 1:** {object: bathtubs, position:(54,71,55)}, {object: toilet, position:(9,91,28)}, {object: sink, position: (5, 74, 39)}, {object: towel, position:(16,105,20)} ...
> **Object Group 2:** {object: door, position:(66,62,59)}, {object: paint, position:(125,27,93)}, {object: table, position:(114,107,15)}, {object: sofa, position:(131,94,22)}, {object: couch, position: (120,125,19)}, {object: desk, position: (159,104,55)}, {object: chair, position: (196,58,34)} ...
> **Object Group 3:** {object: dressing, position:(243,23,29)}, {object: nightstand, position: (288, 54,31)}, {object: bed, position: (274,48,25)} ...
> ... (more groups)

Note that in order to reduce the computation cost of the navigation map, we compute the KV cache of each group only once and store the KV cache in memory. As the robot explores the environment, new objects will be added to different groups, whose KV cache will be updated accordingly.

**2. Memory retrieval**: for a given target, we then leverage semantic-aware memory retrieval to select relevant groups from the navigation map. Then, the KV cache of corresponding groups will be loaded to device memory. Thereby, memory retrieval not only reduces the memory loading cost but also helps the LLM planner to focus on the relevant environment to make better decisions. Let's assume the Group 2 and 3 in the navigation map above are selected.

**3. Planning**: Given the retrieved KV cache, we now query the LLM to plan for the exploration. The prompt for the LLM is as follows:

> **Prompt for Planning**
>
> **Object Group 2:** {object: door, position:(66,62,59)}, {object: paint, position:(125,27,93)}, {object: table, position:(114,107,15)}, {object: sofa, position: (131,94,22)}, {object: couch, position: (120,125,19)}, {object: desk, position: (159,104,55)}, {object: chair, position: (196,58,34)} ...
> **Object Group 3:** {object: dressing, position:(243,23,29)}, {object: nightstand, position: (288, 54,31)}, {object: bed, position: (274,48,25)} ...
> **Instruction:** You are a navigation robot. The above is a description of different objects in the environment that you have seen. Your final goal is to find the TV in the environment. Based on the environmental information, please choose one specific object to travel to as your sub-goal, following such format: "The next subgoal is xxx at position (xx, xx, xx)". Here are the objects that you have traveled to before:
> **Trajectory:** You have visited the door at position (66,62,59) and the dressing at position (243,23,29).

Given the prompt above, the LLM generates the following instruction:

> **Planner Output**
>
> The next subgoal is sofa at position (131,94,22).

The robot will thereby follow the instruction to find the sofa at position (131, 94, 22). The process is repeated until the robot finds the target object.

# D   Network Architecture

We benchmark EfficientNav using four models: LLaVA-7B, LLaMA-3.2-11B, LLaVA-13B, and LLaVA-34B, each comprising an LLM backbone and a ViT-based vision encoder. As detailed in Table 7, their architectures scale systematically.

For LLM Backbones, LLaVA-7b, LLaMA-3.2-11b, LLaVA-13b, and LLaVA-34b adopt Mistral-7B-Instruct-v0.2, LLaMA-3.1, Vicuna-13B-v1.5, and Nous-Hermes-2-Yi-34B, respectively. Both the number of layers and hidden dimensions increase with model scale. For vision encoders, LLaVA

variants utilize a 24-layer ViT with a hidden dimension of 1024, while LLaMA-3.2-11B employs a deeper 32-layer ViT (hidden dimension of 1280). All vision encoders are fine-tuned during training.

Table 7: Network architecture

| Model | LLM backbone | | | Vision Encoder | |
| | LLM | # Layers | Hidden Dim | # Layers | Hidden Dim |
|---|---|---|---|---|---|
| LLaVA-7b | Mistral-7B-Instruct-v0.2 | 32 | 4096 | 24 | 1024 |
| LLaMA-3.2-11b | LLaMA-3.1 | 40 | 4096 | 32 | 1280 |
| LLaVA-13b | Vicuna-13b-v1.5 | 40 | 5120 | 24 | 1024 |
| LLaVA-34b | Nous-Hermes-2-Yi-34B | 60 | 7168 | 24 | 1024 |

# E    CLIP Dependency

In semantics-aware retrieval, we use CLIP to remove redundant information and select relevant groups before sub-goal selection. CLIP is widely used in semantic matching [70, 81, 11]. In fact, as shown in Table 8, other semantic matching models can also be used and show reasonable performance (e.g.,MPNet[57], MiniLM [64]), showing the robustness of our method. While the LLM-based selection method shows

Table 8: Ablation study on group selection method.

| Method | SR | SPL | Real-time Latency |
|---|---|---|---|
| CLIP (ours) | 80.0 | 41.5 | 0.87s |
| MPNet [57] | 79.3 | 39.6 | 0.93s |
| MiniLM [64] | 80.3 | 40.1 | 0.84s |
| LLM | 80.7 | 42.1 | 6.55s |

high real-time latency because of the high computation cost, with no significant success rate improvement. This is because CLIP is enough to remove most of the redundant information and keep the most relevant groups. Even if a little redundant information is kept, the LLM planner can exclude it in the sub-goal selection period. So a light-weight semantic matching model is more reasonable.

# F    Open-vocabulary Evaluation

In Section 4, we follow existing works use the HM3D dataset for comparison. However, we hope to emphasize that EfficientNav is not limited to HM3D and can be applied to open-vocabulary datasets. Here we evaluate our method on HM3D-OVON [78], which has more target object categories. The result is shown in Table 9, and our method also shows strong performance on open-vocabulary setting.

Table 9: Evaluation on HM3D-OVON.

| Method | SR | SPL |
|---|---|---|
| Uni-NaVid [82] | 41.3 | 21.1 |
| FiLM-Nav [77] | 44.9 | 24.5 |
| EfficientNav (ours) | 63.5 | 31.6 |

# G    Implementation Details

In EfficientNav, we use smaller open-source planners, such as LLaVA and LLaMA3.2. We use a single NVIDIA RTX A6000 GPU to deploy LLaVA-7b and LLaMA3.2-11b. When using LLaVA-13b and LLaVA-34b, we deploy our system on 2 and 4 NVIDIA RTX A6000 GPUs, respectively. We also deploy LLaVA-7b on a single Jetson AGX Orin. For the detection model, we use Grounding Dino [37].

# H    Comparison with Other Graph-based Method

Graph-based navigation maps are increasingly favored for their ability to explicitly represent the semantics and locations of visited objects. Among them, HOZ [86] proposes a hierarchical object-to-zone (HOZ) graph to provide coarse-to-fine guidance for navigation robots in unseen environments, which is constructed from scene, zone, and object nodes, updated online based on real-time observations. Based on HOZ, HOZ++ [87] introduces an adaptive graph structure and heterogeneous graph

fusion for the graph to further enhance object navigation performance. LROGNav [60] enhances navigation efficiency by integrating common-sense knowledge of object-to-room relationships extracted from large language models using both positive and negative chain-of-thought promptings, and by training a multi-channel Swin-UNet with multimodal inputs. Imagine Before Go [88] introduces a Self-supervised Generative Map (SGM) and improves exploration efficiency by learning explicit contextual relationships between objects and environments through self-supervised training. It leveraging both episodic observations and general knowledge from Large Language Models to generate unobserved regions of local semantic maps.

Our method EfficientNav improves the efficiency of LLM-based object goal navigation by navigation map caching and retrieval. Different from these works, our method focuses on the computation and management of KV cache for the navigation map description. We use discrete memory caching to decouple the context order and KV cache computation. For object grouping, we use attention-based memory clustering to reduce the impact of ignoring cross-attention. These methods enable KV cache reuse and significantly accelerate the prefilling stage of LLM planning.

## I  Limitations

In this section, we discuss the current limitations and potential avenues for future research.

First, in our method, we propose EfficientNav, enabling efficient zero-shot on-device ObjNAv. In fact, our memory caching and memory retrieval method also benefits the LLM inference on the cloud server by avoiding re-computation and removing redundant information. However, EfficientNav can not solve the problems of high communication latency or privacy concerns that cloud serving faces. Also, in our experiment, EfficientNav achieves $6.7\times$ real-time latency reduction compared with GPT-4 planner. However, as the inference speed of LLM can not reach that of small models, EfficientNav should be carefully used in applications that need extremely low real-time latency.

## J  Broader Impacts

In our paper, our method enables efficient on-device zero-shot indoor ObjNav, overcoming the tight memory constraints by designing novel memory caching and retrieval mechanisms. However, our LLM-based ObjNav works in a zero-shot manner. This can avoid the heavy training cost, but the pre-trained LLM may not master some highly specialized knowledge. Therefore, in high-risk areas such as chemical laboratories with hazardous materials, EfficientNav should be used carefully.

