# OpenReview forum: "EfficientNav: Towards On-Device Object-Goal Navigation with Navigation Map Caching and Retrieval"
_NeurIPS.cc/2025/Conference — NeurIPS 2025 poster_

### Official Review · Reviewer_hd8c · 2025-07-02

**Clarity:** 3
**Significance:** 2
**Originality:** 2
**Rating:** 4
**Confidence:** 3

**Summary:**

This paper presents EfficientNav, a system for enabling efficient zero-shot object-goal navigation (ObjNav) on local devices with limited memory resources. The key challenge addressed is that existing LLM-based navigation systems rely on cloud-based large models like GPT-4, which suffer from high latency and privacy concerns. EfficientNav introduces three main contributions: (1) discrete memory caching to avoid recomputing KV cache for navigation map descriptions, (2) attention-based memory clustering to group related objects and reduce cross-attention loss, and (3) semantics-aware memory retrieval using CLIP to select relevant object groups based on navigation goals. The system achieves 11.1% improvement in success rate over GPT-4 baselines while providing 6.7× real-time latency reduction.

**Questions:**

Questions

Effectiveness of Discrete Memory Caching: Table 4 has shown that the Discrete Memory Caching module achieves 0.4% SR improvement. However, as Section 3.2 shows, this is mainly a design to solve the memory constraints and will cause the ignorance of cross-attention between groups, so how can this module increase the SR metric, or is this just a statistical deviation?

Performance difference between discrete attention and full attention: As the authors put in Section 3.2 and 3.3, the Attention-based Memory Clustering method serves as a complement of the Discrete Memory Caching method to make up for the ignorance of cross-attention between groups. Then why will this discrete attention scheme (Discrete Memory Caching + Attention-based Memory Clustering) outperforms the full attention scheme (LLaVA Planner) by 20.2% in SR as shown in Table 4? The text descriptions in Section 4.3 does not show how this discrete attention could achieve such a performance gain.

The changes of semantic groups: Could the authors provide an example of how the semantic groups are updated as the agent is exploring the scene? And as the paper shows, the newly-added object will be added to existing groups based on the semantic similarity, so will it be a problem that some semantically-related but spatially-distant objects be clustered in a group? If yes, how will this influence the final navigation performance?

**Ethical Concerns:**

["NO or VERY MINOR ethics concerns only"]

**Final Justification:**

Author's clarifications have addressed my concerns, and I am maintaining my positive rating.

**Limitations:**

yes

**Quality:**

2

**Strengths And Weaknesses:**

Strengths:

Important Problem Identification:

This paper addresses a critical challenge in LLM/VLM-based navigation, real-time performance, which is a common bottleneck for many navigation methods. This issue is especially pivotal for enabling the practical deployment of such methods in real-world applications.

Technical Contributions:

1. Discrete Memory Caching: This module effectively addresses memory constraints by clustering navigation map information into discrete groups, independently calculating their KV cache, and efficiently reusing cached data to reduce recomputation and transfer overhead.

2. Attention-based Memory Clustering: This approach leverages LLM attention mechanisms to cluster semantically related map information, thereby significantly improving the effectiveness of memory retrieval and reducing the adverse impact of ignoring cross-attention.

3. Semantics-aware Memory Retrieval: By utilizing semantic similarity via the CLIP model, this module dynamically selects relevant map groups based on the navigation task, enabling efficient memory usage and significantly enhancing planning performance of smaller LLMs.

Thorough analysis of memory and efficiency of different LLM/VLMs: This paper clearly presents the memory requirements of different LLM/VLMs and their latency in Figure 1. It also evaluates the latency of different navigation stages in Figure 3 (b) and Figure 9. This thorough evaluation is valuable as a guidance to the design of LLM/VLM-based navigation methods.

Evident performance improvement: The paper provides extensive experiments on the HM3D benchmark, demonstrating significant improvements over existing methods, notably 11.1% success rate improvements over GPT-4-based baselines and substantial latency reductions.

Weaknesses

Attention-Based Clustering Justification: The choice of using only 1/10 of LLM layers for attention-based clustering lacks theoretical justification. Why this specific fraction? How sensitive is performance to this hyperparameter?

CLIP Dependency: The semantics-aware retrieval relies heavily on CLIP's semantic understanding. The paper doesn't analyze failure cases where CLIP's similarity scores might be misleading for navigation decisions. The contribution of CLIP to the final navigation success rates is not discussed. Is this key to the 16.7% success improvement? If not using CLIP or using other similar models, how much will the performance drop?

Limited Baseline Comparisons: This paper primarily compares against naive LLM planners and GPT-4. Comparisons with other efficient LLM inference methods would be valuable.

Practical deployment problems: The practical deployment complexities of integrating multiple sophisticated methods might require considerable development effort for real-world scenarios.

Lack of detailed ablation studies: The ablation study does not show the influence and detailed analysis of the ablation choices, such as the influence of CLIP model, the choice of 1/10 layers of LLMs and how the discrete attention scheme outperforms the full attention scheme.

---

> ### Author Rebuttal · Authors · 2025-07-29
>
> 1 **Ablations for attention-based clustering:**  In practice, we find that the success rate is not very sensitive to this hyperparameter. **We evaluate different choices and find the results reasonable, except for using only 1 layer.** As the attention distribution is relatively uniform in the first layers. We set this threshold of using 1/10 layers to make a trade-off between latency and success rate.
> | Number of Layers |   SR   |  SPL   | Real-time Latency |
> |:----------------:|:------:|:------:|:-----------------:|
> | 1                | 70.0   | 34.8   | 0.82s             |
> | 2                | 78.7   | 41.0   | 0.84s             |
> | 4                | 79.7   | 41.5   | 0.87s             |
> | 6 (1/10 layer)| 80.0 | 41.5 | 0.87s         |
> | 8                | 79.3   | 40.6   | 0.87s             |
>
> 2 **CLIP Dependency:** The core idea of semantics-aware retrieval is that **we need to remove redundant information before giving the environment description to the LLM planner, thus leading to success rate improvement** (Section 3.1 and 3.4). Here, we select CLIP to remove redundant information and select relevant groups because CLIP is widely used in semantic matching [1, 2, 3]. In fact, **other semantic matching models can also be used and show reasonable performance (e.g.,all-mpnet-base-v2, all-MiniLM-L6-v2), showing the robustness of our method.** We also compared our method with the position-based method (select groups according to average object distance) and the LLM-based method (remove redundant information and select final sub-goal both using LLM). For the position-based method, we evaluate both considering/not considering device memory capacity in memory retrieval, which is discussed in Line 223. Position-based method shows a low success rate as it does not consider semantics. And the LLM-based method shows high real-time latency because of the high computation cost, with no significant success rate improvement. This is because CLIP is enough to remove most of the redundant information and keep the most relevant groups. Even if a little redundant information is kept, the LLM planner can exclude them in sub-goal selection. So a light-weight semantic matching model is important.
> | Method                                      |   SR   |  SPL   | Real-time Latency |
> |:-------------------------------------------:|:------:|:------:|:-----------------:|
> | position-based (without considering device memory capacity in memory retrieval)    | 63.3   | 34.2   | 2.32s           |
> | position-based (considering device memory capacity in memory retrieval)  | 47.5   | 25.8   | 0.81s           |
> | **CLIP (ours)**                             | 80.0 | 41.5 | 0.87s      |
> | all-mpnet-base-v2                           | 79.3   | 39.6   | 0.93s         |
> | all-MiniLM-L6-v2                            | 80.3   | 40.1   | 0.84s         |
> | LLM                                         | 80.7   | 42.1   | 6.55s        |
>
> 3 **Evaluation on efficient LLM inference methods:**
> Our discrete memory caching method is similar to sparse attention methods. As there is no existing sparse attention method that focuses on navigation, we compare with general adaptive sparse attention methods.
> **There is a core difference between general adaptive sparse attention and EfficientNav:** EfficientNav clusters objects into different groups and calculates the attention of each group individually. **This structured sparse approach can decouple the group order and KV cache computation of each group, thus enabling KV reuse when we retrieve different groups in navigation planning.** However, for general adaptive sparse attention methods, **their purpose is to minimize the difference between sparse attention and full attention**, and accelerate the attention calculation. But without memory retrieval, even full attention shows a low success rate, as discussed in Section 3.1. At the same time, without discrete memory caching, these methods need to recompute the KV cache of the navigation map description because of map changes, thus showing longer real-time latency.
>
> We conduct experiments to compare these sparse attention works with our method. Our method outperforms all the baselines. We will cite all these papers and compare them with EfficientNav in our final version.
> | Method        |   SR   |  SPL   | Real-time latency |
> |:-------------:|:------:|:------:|:----------------:|
> | Minferenc     |  35.3  |  18.0  |3.88s|
> | FlexPrefill   |  36.7  |  20.1  |3.02s|
> | EfficientNav  |  80.0  |  41.5  |    0.87s       |
>
> 4 **Lack of real-world deployment:**
> Although we did not evaluate on a real-world robot, **we deployed our model on the computation platform (Jetson Orin) that is frequently used by real-world robots**, as our method mainly solves the problem of high computation cost in prefilling by optimizing KV cache management.
> And we would like to clarify that **simulation-based evaluation is also a widely used practice in object-goal navigation research, as seen in many prior works (e.g., [4, 5, 6]).** Our experimental setup follows the same standard, ensuring fair comparison with existing methods while maintaining reproducibility.
>
> 5 **Effectiveness of Discrete Memory Caching**: only using discrete memory caching will not affect the success rate much, and the SR improvement is just a statistical deviation.
>
> 6 **Effectiveness Attention-based memory Clustering:** the planning performance is highly related to the objects selected for the planner. And **the object selection is related to (1) how to cluster objects and (2) how to select groups.**
>
>  As discussed in Line 322, in LLaVA planner baseline, we also cluster objects and select groups based on position, following [4, 7] (when not using selection, the success rate is only 36.7% caused by the redundant information). However, as the selected objects have poor correlation with each other, the success rate is still low whether using full attention or discrete attention. **After attention-based memory clustering is used, objects in each selected group have high correlation, which helps the LLM to better understand the environment and reduces the impact of ignoring cross-attention at the same time, thus improving the success rate.** When using semantics-aware memory retrieval, we can select the most related groups, which further improves the success rate. And our semantic-aware memory retrieval considers memory capacity, which is not considered by [4, 7], thus showing lower real-time latency. We will add the detailed explanation in our final version.
>
> 7 **The changes of groups:** We conduct object clustering based on **both semantics and position**. As discussed in Section 3.3, we use the **LLM attention distribution** to cluster the groups. When new objects are detected, we input the information of existing groups and the newly detected objects into LLM. The information contains both semantics and positions of the groups/objects. And we infer a few layers of the LLM and cluster object based on the attention score between newly detected objects and existing groups. This can reduce the difference between discrete attention and full attention. In conclusion, **we use the attention of LLM itself to cluster the newly detected objects into different groups, both considering semantics and positions.** We also evaluate clustering only based on position (cluster the new object into the nearest group) and semantics (use CLIP to calculate the similarity between new objects and existing groups), and we find our attention-based method achieves the best performance.
> | Clustering Method  |   SR   |  SPL   |
> |:------------------:|:------:|:------:|
> | position-only      | 45.3   | 23.4   |
> | semantic-only      |  70.8 | 34.6     |
> | **attention-based** | 80.0 | 41.5 |
>
> It is true that some semantically related but spatially distant objects can be clustered in a group, e.g., objects in two different bathrooms. However, **this is reasonable as these objects are usually both relevant/irrelevant to the current task.** In the planning process, both the position and semantics of each selected object will be given to the LLM, and the LLM will analyze the environment and choose only one object as sub-goal.
> **We will add a figure to show an example of grouping updating in our final version, as adding figures in rebuttal is not allowed.**
>
> [1]Wu, Xiaoshi, et al. "Cora: Adapting clip for open-vocabulary detection with region prompting and anchor pre-matching."
>
> [2]Zhang, Beichen, et al. "Long-clip: Unlocking the long-text capability of clip."
>
> [3]Dorbala, Vishnu Sashank, et al. "Clip-nav: Using clip for zero-shot vision-and-language navigation."
>
> [4] Shah, Dhruv, et al. "Navigation with large language models: Semantic guesswork as a heuristic for planning."
>
> [5] Long, Yuxing, et al. "Instructnav: Zero-shot system for generic instruction navigation in unexplored environment."
>
> [6] Wu, Pengying, et al. "Voronav: Voronoi-based zero-shot object navigation with large language model."
>
> [7]Zhang, Junbo, and Kaisheng Ma. "MG-VLN: Benchmarking Multi-Goal and Long-Horizon Vision-Language Navigation with Language Enhanced Memory Map."

---

> > ### Comment · Reviewer_hd8c · 2025-08-06
> >
> > Thank you for the thorough rebuttal and the additional experimental results. Your clarifications have addressed my concerns, and I am maintaining my positive rating.

---

> > > ### Author Response · Authors · 2025-08-07
> > > **Thank you for your positive feedback**
> > >
> > > Thank you for your positive feedback. We are glad our rebuttal can resolve your concerns, and we will carefully address all your suggestions in the final version to further improve the paper.

---

### Official Review · Reviewer_LqYc · 2025-07-02

**Clarity:** 3
**Significance:** 3
**Originality:** 3
**Rating:** 5
**Confidence:** 4

**Summary:**

The authors propose discrete memory caching for object-goal navigation, whereby the map information ca split into segments and the KV cache can be more efficiently decomposed and stored. This solution mitigates the problem of constrained systems not having enough memory to store entire KV caches. Since this clustering into discrete segments may negatively impact performance due to problems with cross-attention across segments, the authors propose attention-based memory clustering.

**Questions:**

For semantic retrieval, do you think something like a knowledge graph or a scene graph can be used to augment or replace your current CLIP-based method?

**Ethical Concerns:**

["NO or VERY MINOR ethics concerns only"]

**Final Justification:**

The paper is very interesting and could enable low-SWaP or edge navigation in cloudless environments through the proposed KV caching. I am content with this paper being accepted.

**Limitations:**

Yes

**Quality:**

3

**Strengths And Weaknesses:**

Strengths:
* The authors include a nice empirical validation (Figure 6, a) that the latency incurred by loading the KV cache repeatedly in the conventional way is large.
* Semantics-aware memory retrieval is used by leveraging a CLIP to compute semantically similar inputs, which seems reasonable.
* Their experimental results on improving latency over the state-of-the-art are impressive.

Weaknesses:
* Some well-known related work on navigation using LLMs is missing, including the line of work on integrating scene graphs and LLMs for navigation, see, e.g., [1] below.
[1] Rajvanshi, Abhinav, et al. "Saynav: Grounding large language models for dynamic planning to navigation in new environments." Proceedings of the International Conference on Automated Planning and Scheduling. Vol. 34. 2024.

---

> ### Author Rebuttal · Authors · 2025-07-29
>
> 1 Related work referring: we thank the reviewer for their valuable suggestion to refer to these foundational works.
> **SayNav** novelly uses a grounding mechanism, building a 3D scene graph of the explored environment as input to LLMs, for generating feasible and contextually appropriate high-level plans for navigation. Our method **EfficientNav** improves the efficiency of LLM-based object goal navigation by navigation map caching and retrieval. Different from SayNav, our method focuses on the computation and management of KV cache for the navigation map description. We use discrete memory caching to decouple the context order and KV cache computation, which enables KV cache reuse. And we also use attention-based memory clustering to reduce the impact of ignoring cross-attention. We will cite these papers and compare them with EfficientNav in our final version.
>
>
> 2 Retrieval using knowledge graph: **a knowledge graph can be used to augment our method, while our CLIP-based method is also necessary.**  Existing works attempt to use a knowledge graph to embed prior knowledge, such as object relationships, to guide the navigation in unseen environments. In the navigation process, a knowledge graph can provide correlated objects as auxiliary information to locate the target object [1, 2]. However, **our CLIP-based method is still necessary in the following aspects**:
>
> (1) Better generalization: knowledge graph requires manually curated or statistically learned object co-occurrence priors. And the navigation performance is highly related to the built knowledge graph, limiting generalization to unseen objects/scenes. Some works combine knowledge graphs with online learning, while the graph is still based on a fixed initial graph [3].
> While our CLIP-based method can enable zero-shot matching after large-scale pre-training. And our navigation map is totally built from scratch, which can better adapt our method to new scenes.
>
> (2) High flexibility in similarity computation: in existing works, the knowledge graph is mainly used to evaluate the relationship between different objects. However, the CLIP-based method has higher flexibility, which can both evaluate object-object similarity and object-group similarity. Especially, the group is dynamic in our method, which further shows the flexibility of the CLIP-based method.
>
> At the same time, **Knowledge Graph Methods have their own strength**:
>
> (1) Better efficiency: CLIP requires text encoding and similarity computation, which will introduce extra latency. While a knowledge graph only queries a precomputed graph. **However, in indoor navigation scenes, the number of groups and objects is not too large, and the latency caused by CLIP is not the bottleneck.** So we use a CLIP-based method to pursue a higher success rate. We will consider using a knowledge graph in our future work for larger-scale navigation. Thank you for your insightful comments.
>
> [1]Yang, Wei, et al. "Visual semantic navigation using scene priors."
>
> [2]Xu, Nuo, et al. "Aligning knowledge graph with visual perception for object-goal navigation."
>
> [3]Zhang, Sixian, et al. "Hierarchical object-to-zone graph for object navigation."

---

> > ### Comment · Reviewer_LqYc · 2025-08-04
> > **Response**
> >
> > Thank you for the detailed response. I appreciate the nuance you mention between CLIP-based semantic matching and using knowledge or scene graphs. It makes sense that CLIP is not necessarily a bottleneck. The paper is very interesting and could enable low-SWaP or edge navigation in cloudless environments through the propose KV caching.

---

> > > ### Author Response · Authors · 2025-08-06
> > > **Thank you for your positive feedback**
> > >
> > > Thank you for your positive feedback and we will carefully address your suggestions in the final version to further improve the paper. We thank you again for your insightful comments.

---

### Official Review · Reviewer_QLLB · 2025-07-03

**Clarity:** 2
**Significance:** 3
**Originality:** 3
**Rating:** 4
**Confidence:** 4

**Summary:**

EfficientNav is proposed to address two major challenges in applying large language models (LLMs) to on-device Object Navigation (ObjectNav). The first challenge lies in the difficulty of deploying large LLMs on resource-constrained devices. The second issue concerns the growth of the key-value (KV) cache, which increases significantly as context information accumulates throughout the navigation process. To tackle these challenges, the authors introduce a combination of discrete memory caching and a cluster-based navigation map, allowing for more efficient estimation and control of KV cache usage. Additionally, they propose a semantic-aware memory retrieval module, which enables the system to reduce unnecessary map information by selecting only semantically relevant memory entries for reasoning and decision-making.

**Questions:**

1. What is the basis for clustering groups? Is it based on semantics or other metrics?

2.  In the supplementary navigation examples, does each node represent only one object?
In Figure 11, object 7 and object 10 appear to belong to the same node, but in the trajectory, both objects appear separately. This seems contradictory.

3. How is the sub-goal selected?
Section 3.4 describes how relevant groups are selected, but since each group contains multiple objects, how is the final sub-goal selected from within a group?

**Ethical Concerns:**

["NO or VERY MINOR ethics concerns only"]

**Final Justification:**

This paper introduces an efficient navigation model for indoor navigation and I recommend this paper for borderline accept.

**Limitations:**

- The paper lacks detailed architectural descriptions, particularly regarding components such as the object detection module. This omission makes it difficult to fully understand or reproduce the system pipeline.
- The empirical evaluation is limited to indoor simulation environments. There is little analysis of how well the proposed method generalizes to more complex, real-world, or large-scale navigation scenarios.

**Paper Formatting Concerns:**

None.

**Quality:**

3

**Strengths And Weaknesses:**

### Strengths
- The proposed Attention-based Memory Clustering and Semantic-aware Memory Retrieval effectively enhance the usability of LLMs for on-device navigation tasks.
- These two components enable the model to identify relevant features based on their relation to the current sub-goal, improving both retrieval efficiency and navigation accuracy.

### Weaknesses
- The overall architecture of the model is not fully described in the paper. And make the reader a little bit confusing. For example, it does not state which detection model is used for constructing navigation map.

---

> ### Author Rebuttal · Authors · 2025-07-29
>
> 1 **Model architecture confusion:** We use **grounding-dino** as our detection model.
> **While the choice of detection model impacts the semantic map's accuracy, it does not affect the core contribution or the primary problem addressed in this paper.**
> Here we choose 3 detection models to evaluate our system, and the final success rate does not change much. Note that **we mainly focus on the planning process of object navigation, which represents the primary efficiency bottleneck, rather than the perception process.** We will clearly explain this in our final version.
> | Detection Model  |   SR   |  SPL   |
> |:----------------:|:------:|:------:|
> | Grounding-dino   |  80.0  |  41.5  |
> | OWL-ViT         |  80.3  |  41.2  |
> | Detic           |  79.0  |  41.1  |
>
> Here we re-explain the working process of our navigation system. First, as shown in Figure 2, using detection models in the navigation process, we can construct a navigation map and add newly detected objects into the map in each navigation step.  The navigation map records the semantics and position of each object. Second, as shown in Figure 5, we use attention-based memory clustering to cluster each newly detected object into related groups. This can reduce the impact of ignoring cross-attention between different groups. Third, also shown in Figure 5, we retrieve groups related to the current goal object, and feed the information of these groups into the LLM planner as prompt. Then, we load the KV cache of related groups calculated by discrete memory caching to avoid re-computation, and the KV cache of newly detected objects in retrieved groups is calculated. Finally, the LLM planner generates the next sub-goal.
>
> 2 **Basis for clustering groups:** As discussed in Section 3.3, we use the **LLM attention distribution** to cluster the groups. When new objects are detected, we input the information of existing groups and the newly detected objects into LLM. The information contains **both semantics and positions** of the groups/objects. And we infer a few layers of the LLM and cluster object based on the attention score between newly detected objects and existing groups. This can reduce the impact of ignoring cross-attention between groups. In conclusion, we use the attention of LLM itself to cluster the newly detected objects into different groups, both considering semantics and positions. We also evaluate clustering only based on position (cluster the new object into the nearest group) and semantics (use CLIP to calculate the similarity between new objects and existing groups), and we find our attention-based method achieves the best performance.
> | Method              |   SR   |  SPL   |
> |:-------------------:|:------:|:------:|
> | position-based      |  45.3  |  23.4  |
> | Semantic-based      |  70.8 | 34.6  |
> | **Attention-based (ours)** |  80.0  |  41.5  |
>
> 3 **Node represents:** in this figure, one node **does not** represent only one object. To be specific, multiple objects related to one node only means that these objects are near the same place. It does not mean all these objects are at the center of the node. Instead, there is some distance between them, so we can visit different objects related to the same node separately. Many existing works use a similar representation manner [3, 4, 5, 7].
>
> 4 **The way sub-goal is selected:** the final sub-goal is selected **by the LLM planner**. To be specific, **the sub-goal is determined in a two-step manner**: first, we select the relevant groups. Second, we select a sub-goal in the relevant groups. Both steps are important because if we directly select the final sub-goal among all groups, the planning performance will drop because of the huge amount of redundant information, as discussed in Section 3.1.
>
> For the first step, we use CLIP to select relevant groups for better efficiency, as the task of removing redundant information is much easier than finding a specific sub-goal. For the second step, we use LLM to select the final sub-goal among the relevant groups. We also evaluate the result when using CLIP or LLM in both steps. When both are using CLIP, the accuracy drops a lot, as CLIP has a lower reasoning ability than LLM. When both are using LLM, the real-time latency becomes very long, as LLM shows higher inference latency than CLIP. And there is no significant success rate gain, as the CLIP model is enough to select relevant groups.
> | Method          |   SR   |        SPL       | Real-time Latency |
> |:---------------:|:------:|:----------------:|:-----------------:|
> | CLIP+CLIP       | 5.3  | 1.1            | 0.19s             |
> | LLM+LLM         | 80.7   | 42.1             | 6.55s            |
> | **CLIP+LLM (ours)** | 80.0   | 41.5        | 0.87s        |
>
> 5 **Evaluation environment limited**: we would like to clarify that **EfficientNav is specially designed for indoor navigation.** We highlight that **indoor navigation is an important research area,** and is adopted in many applications, such as household robots and healthcare robots. **Many existing works also focus on indoor navigation** [1, 2, 3, 4, 5, 6, 7].
> And **simulation-based evaluation is also a widely used practice in object-goal navigation research**, as seen in many prior works (e.g., [1, 2, 3, 4, 5, 6, 7]). Our experimental setup follows the same standard, ensuring fair comparison with existing methods while maintaining reproducibility.
>
> [1] Shah, Dhruv, et al. "Navigation with large language models: Semantic guesswork as a heuristic for planning."
>
> [2] Long, Yuxing, et al. "Instructnav: Zero-shot system for generic instruction navigation in unexplored environment."
>
> [3] Wu, Pengying, et al. "Voronav: Voronoi-based zero-shot object navigation with large language model."
>
> [4]Chen, Jiaqi, et al. "Mapgpt: Map-guided prompting with adaptive path planning for vision-and-language navigation."
>
> [5]Zhang, Sixian, et al. "Hierarchical object-to-zone graph for object navigation."
>
> [6]Sun, Leyuan, et al. "Leveraging Large Language Model-based Room-Object Relationships Knowledge for Enhancing Multimodal-Input Object Goal Navigation."
>
> [7]Zhou, Gengze, Yicong Hong, and Qi Wu. "Navgpt: Explicit reasoning in vision-and-language navigation with large language models."

---

> ### Comment · Reviewer_QLLB · 2025-08-06
> **Response**
>
> - Thank you to clarify the model architecture and some details of model design, experiments.
> - I will raise score to BA in this time.

---

> > ### Author Response · Authors · 2025-08-06
> > **Thank you for your positive feedback**
> >
> > Thank you for your positive feedback. We are glad our rebuttal can resolve your concerns, and we will carefully address all your suggestions in the final version to further improve the paper.

---

### Official Review · Reviewer_gaGt · 2025-07-03

**Clarity:** 2
**Significance:** 2
**Originality:** 3
**Rating:** 4
**Confidence:** 3

**Summary:**

This paper presents an efficient LLM based object goal navigation approach with navigation map cashing and retrieval.

**Questions:**

See Weakness

Section 3.3 and 3.4 is a little difficult to grasp. It would be nicer to use Figure 5 and tell how LLM semantic space and 2D map is associated.

**Ethical Concerns:**

["NO or VERY MINOR ethics concerns only"]

**Final Justification:**

While the proposed framework may not be on-the-fly deployable on real robots, most of my concerns are resolved and I found its attention based clustering interesting.

**Limitations:**

yes

**Quality:**

3

**Strengths And Weaknesses:**

Strength:

- The motivation toward on device LLM based object goal navigation is valuable.

- The proposed method achieves higher success rate than baseline models with relatively low latency.


Weakness:

1) The task setting is specialized: the goal object is defined only three. And it does not do vision language navigation which is tackled by its LLM baseline techniques such as MapGPT. Evaluating on object goal navigation with the same number of target object as in previous works and/or VLN would be appreciated.

2) It talks about Jetson but there is no robot hardware deployment experiment.

3) I wonder how the general adaptive sparse attention approaches work on the object goal navigation task, since its not predefined so it could in theory adapt to this kind of task too. Comparisons to the model using dynamic sparse attention like these without the proposed semantics aware memory retrieval would be appreciated.

- Minference

- FlexPrefill: A Context-Aware Sparse Attention Mechanism for Efficient Long-Sequence Inference

- SpargeAttention: Accurate and Training-free Sparse Attention Accelerating Any Model Inference

4) It lacks some object goal navigation references that relate target and zone/room.

- Hierarchical Object-to-Zone Graph for Object Navigation

- Enhancing Multimodal-Input Object Goal Navigation by Leveraging Large Language Models for Inferring Room-Object Relationship Knowledge

- HOZ++: Versatile Hierarchical Object-to-Zone Graph for Object Navigation

---

> ### Author Rebuttal · Authors · 2025-07-29
>
> 1.1 **Limited number of goal objects:** we appreciate the reviewer's attention to our experimental setup. However, we would like to clarify that our work actually **evaluates six object categories** (toilet, TV, chair, sofa, bed, and plant), strictly following the setting in the HM3D dataset (in our paper, we only show three of the goals at Line 243 as an example). In addition, to show the robustness of our method, we show the success rate of each individual object goal here. **As our method is actually open-vocabulary, we further evaluate 4 more object goals (table, door, sink, lamp) to show our generalization.** The result shows that our method achieves a high success rate on all object goals.
> | Object  | Toilet | TV    | Chair  | Sofa  | bed   | plant | table | door | sink  | lamp  |
> |---------|--------|-------|--------|-------|-------|-------|-------|------|-------|-------|
> | SR      | 72.5%  | 75.0% | 87.5%  | 90.0% | 85.0% | 70.0% | 95.0% | 100% | 72.5% | 82.5% |
>
> 1.2 **MapGPT evaluation:** we also evaluate MapNav on the HM3D ObjNav dataset. According to the following table, our method achieves 14.5% SR and 6.3% SPL improvements over MapGPT, as our method removes redundant information in the prompt using semantics-aware memory retrieval. We also achieve 7.0x real-time latency reduction by using memory caching and retrieval.
> | Method        |   SR   |  SPL   | Real-time latency |
> |:-------------:|:------:|:------:|:----------------:|
> | MapGPT        |  65.5  |  35.2  |      6.08s       |
> | EfficientNav  |  80.0  |  41.5  |      0.87s       |
>
> 2 **Lack of robot deployment:** We utilize a Jetson Orin as the compute device, a common choice for on-board computing hardware in robotics, and conduct experiments within the Habitat simulator. Specifically, the simulator runs on a separate computer, sending sensor data to the Jetson Orin, which hosts the LLM planner. The Jetson Orin then generates the plan and transmits it back to the separate computer, forming a closed-loop system. We would like to clarify that **simulation-based evaluation is a widely used practice** in object-goal navigation research, as evidenced by many prior works (e.g., [1, 2, 3]). Our experimental setup follows the same standard, ensuring fair comparison with existing methods while maintaining reproducibility.
>
> 3 **Evaluation on sparse attention approaches:** the goals of general adaptive sparse attention and EfficientNav are different: EfficientNav clusters objects into different groups and calculates the attention of each group individually. **This structured sparse approach can decouple the group order and KV cache computation of each group, thus enabling KV reuse when we retrieve different groups in navigation planning.** However, for general adaptive sparse attention methods, **their purpose is to minimize the difference between sparse attention and full attention** and accelerate the attention calculation. But without memory retrieval, even full attention shows a low success rate, as discussed in Section 3.1. At the same time, without discrete memory caching, these methods need to recompute the KV cache of the navigation map description because of map changes, thus showing longer real-time latency.
>
> We conduct experiments to compare these sparse attention works with our method. Our method outperforms all the baselines. We will cite all these papers and compare them with EfficientNav in our final version.
> | Method        |   SR   |  SPL   | Real-time latency |
> |:-------------:|:------:|:------:|:----------------:|
> | Minferenc     |  35.3  |  18.0  |3.88s|
> | FlexPrefill   |  36.7  |  20.1  |3.02s|
> | EfficientNav  |  80.0  |  41.5  |    0.87s       |
>
> 4 **Lack of references**: we thank the reviewer for their valuable suggestion to refer to these foundational works.
>
> **HOZ** innovatively proposes a hierarchical object-to-zone (HOZ) graph to provide coarse-to-fine guidance for navigation robots in unseen environments, which is constructed from scene, zone, and object nodes, updated online based on real-time observations.
>
> Based on HOZ, **HOZ++** cleverly introduces an adaptive graph structure and heterogeneous graph fusion for the graph to further enhance object navigation performance.
>
> **LROGNav** enhances navigation efficiency by integrating common-sense knowledge of object-to-room relationships extracted from large language models using both positive and negative chain-of-thought promptings, and by training a multi-channel Swin-UNet with multimodal inputs.
>
> We also find a paper called **Imagine Before Go**, which introduces a Self-supervised Generative Map (SGM) and improves exploration efficiency by learning explicit contextual relationships between objects and environments through self-supervised training. It leveraging both episodic observations and general knowledge from Large Language Models to generate unobserved regions of local semantic maps.
>
> Our method **EfficientNav** improves the efficiency of LLM-based object goal navigation by navigation map caching and retrieval. Different from these works, **our method focuses on the computation and management of KV cache for the navigation map description.** We use discrete memory caching to decouple the context order and KV cache computation. For object grouping, we use attention-based memory clustering to reduce the impact of ignoring cross-attention. These methods enable KV cache reuse and significantly accelerate the prefilling stage of LLM planning.
>
> We will cite all these papers and compare them with EfficientNav in our final version.
>
> 5 **Writing unclarity**: we apologize for the writing unclarity. Here we re-explain how the LLM semantic space and 2D map are associated. First, as shown in Figure 2, using detection models in the navigation process, we can construct a navigation map and add newly detected objects into the map in each navigation step.  The navigation map records the semantics and position of each object. Second, as shown in Figure 5, we use attention-based memory clustering to cluster each newly detected object into related groups. This can reduce the impact of ignoring cross-attention between different groups. Note that the clustering process is based on the LLM attention, which is not directly determined by the object position on the map. Third, also shown in Figure 5, we retrieve groups related to the current goal object, and feed the information of these groups into the LLM planner as prompt. Then, we load the KV cache of related groups calculated by discrete memory caching to avoid re-computation, and the KV cache of newly detected objects in retrieved groups is calculated. Finally, the LLM planner generates the next sub-goal.
> We will fix our writing in the final version.
>
> [1] Shah, Dhruv, et al. "Navigation with large language models: Semantic guesswork as a heuristic for planning."
>
> [2] Long, Yuxing, et al. "Instructnav: Zero-shot system for generic instruction navigation in unexplored environment."
>
> [3] Wu, Pengying, et al. "Voronav: Voronoi-based zero-shot object navigation with large language model."

---

> > ### Comment · Reviewer_gaGt · 2025-08-05
> > **Response**
> >
> > - Thank you for your answer. I understand that the task is object goal navigation on HM3D following the same protocol. The task definition in 4.1 is abit confusing, I'd suggest to modify and elaborate there. I believe it is also not an open vocaburaly setting like HM3D-OVON.
> > - Thanks for clarification on 3.3 and 3.4. I still feel what is stored in the memory or what is input tokens to the LLMs planner are unclear. I suspect it's in the form of like Fig 1 top right.

---

> > > ### Author Response · Authors · 2025-08-05
> > > **Explaination for dataset usage and memory/prompt organization**
> > >
> > > 1 **Dataset usage**: thank you for your valuable suggestion. The HM3D dataset is indeed not an open-vocabulary dataset.  In our paper, we follow existing works [1, 2, 3] to use the HM3D dataset for comparison. **However, we hope to emphasize that EfficientNav is not limited to HM3D and can be applied to open-vocabulary datasets.** We will try our best to evaluate our method on HM3D-OVON. And if we can get the result in time, we will immediately update this comment. If not, we will evaluate our method on HM3D-OVON in the final version if our paper is accepted. We will also fix the task definition in 4.1 and clearly explain the dataset settings in the final version.
> > >
> > > [1] Shah, Dhruv, et al. "Navigation with large language models: Semantic guesswork as a heuristic for planning."
> > >
> > > [2] Long, Yuxing, et al. "Instructnav: Zero-shot system for generic instruction navigation in unexplored environment."
> > >
> > > [3] Wu, Pengying, et al. "Voronav: Voronoi-based zero-shot object navigation with large language model."
> > >
> > > 2 **Unclear memory and prompt**: in Figure 11 in Appendix B, we have shown a simplified example for the EfficientNav system. We now further provide a more detailed example to illustrate the memory and prompt for EfficientNav.
> > >
> > > **Navigation Map Description**: the navigation map records the objects and their position. To facilitate memory retrieval, we organize these objects into groups. Hence, an example of the navigation map is as follows:
> > > ```
> > > Object Group 1: {object: bathtubs, position:(54,71,55)}, {object: toilet, position:(9,91,28)}, {object: sink, position: (5, 74, 39)}, {object: towel, position:(16,105,20)} ...
> > >
> > > Object Group 2: {object: door, position:(66,62,59)}, {object: paint, position:(125,27,93)}, {object: table, position:(114,107,15)}, {object: sofa, position: (131,94,22)}, {object: couch, position: (120,125,19)},{object: desk, position: (159,104,55)}, {object: chair, position: (196,58,34)} ...
> > >
> > > Object Group 3: {object: dressing, position:(243,23,29)}, {object: nightstand, position: (288, 54,31)}, {object: bed, position: (274,48,25)} ...
> > >
> > > ... (more groups)
> > > ```
> > > Note that in order to reduce the computation cost of the navigation map, we compute the KV cache of each group only once and store the KV cache in memory. As the robot explores the environment, new objects will be added to different groups, whose KV cache will be updated accordingly.
> > >
> > > **Memory retrieval**: for a given target, we then leverage semantic-aware memory retrieval to select relevant groups from the navigation map. Then, the KV cache of corresponding groups will be loaded to device memory. Thereby, memory retrieval not only reduces the memory loading cost but also helps the LLM planner to focus on the relevant environment to make better decisions. Let's assume the Group 2 and 3 in the navigation map above are selected.
> > >
> > > **Planning**: Given the retrieved KV cache, we now query the LLM to plan for the exploration. The prompt for the LLM is as follows:
> > > ```
> > > Object Group 2: {object: door, position:(66,62,59)}, {object: paint, position:(125,27,93)}, {object: table, position:(114,107,15)}, {object: sofa, position: (131,94,22)}, {object: couch, position: (120,125,19)},{object: desk, position: (159,104,55)}, {object: chair, position: (196,58,34)} ...
> > >
> > > Object Group 3: {object: dressing, position:(243,23,29)}, {object: nightstand, position: (288, 54,31)}, {object: bed, position: (274,48,25)} ...
> > >
> > > You are a navigation robot. The above is a description of different objects in the environment that you have seen. Your final goal is to find the TV in the environment. Based on the environmental information, please choose one specific object to travel to as your sub-goal, following such format: "The next subgoal is xxx at position (xx, xx, xx)". Here are the objects that you have traveled to before:
> > >
> > > You have visited the door at position (66,62,59) and the dressing at position (243,23,29).
> > > ```
> > > Given the prompt above, the LLM generates the following instruction:
> > > ```
> > > The next subgoal is sofa at position (131,94,22).
> > > ```
> > > The robot will thereby follow the instruction to find the sofa at position (131, 94, 22). The process is repeated until the robot finds the target object.
> > >
> > > We really appreciate your feedback, and we will elaborate on the example to add a more detailed description of the overall workflow in our final version.

---

> > > > ### Comment · Reviewer_gaGt · 2025-08-05
> > > > **Response**
> > > >
> > > > Thanks again for explanations. Although real-world robot experiments are not provided, most of my concerns are clarified and I lean toward positive side at this point.

---

> > > > > ### Author Response · Authors · 2025-08-05
> > > > > **Thank you for your feedback and suggestions**
> > > > >
> > > > > Thank you for your positive feedback, and we will carefully address all your suggestions in the final version to further improve the paper.
> > > > >
> > > > > We are glad our rebuttal can resolve your concerns, and we would greatly appreciate if you can kindly consider raising your score, which can help our paper a lot.

---

> > > > > > ### Comment · Reviewer_gaGt · 2025-08-06
> > > > > > **Response**
> > > > > >
> > > > > > I'd raise my score to BA at this point.
> > > > > > Finally, it would be great to know how the knowledge about the object positions is obtained:
> > > > > > - these object positions are  defined in world frame?
> > > > > > - how are they obtained in real world?

---

> > > > > > > ### Author Response · Authors · 2025-08-06
> > > > > > > **Explaination for object position**
> > > > > > >
> > > > > > > The object positions are defined in the world frame.
> > > > > > >
> > > > > > > The world coordinate system is established with its origin at the robot's initial placement position. During navigation, the robot's pose within the world frame is tracked using odometry. Object positions in the robot frame are determined by combining perception results with depth information from sensors, such as an RGB-D camera. Finally, the objects' positions in the world frame are calculated through a coordinate transformation that uses the robot's position in the world frame and the objects' positions in the robot frame.
> > > > > > >
> > > > > > > To minimize the gap between reality and simulation, several practical engineering methods can be used, such as employing more accurate sensors or developing advanced SLAM and 3D perception algorithms. These tools, when used alongside our proposed approach, can optimize the robot's real-world performance.
> > > > > > >
> > > > > > > Finally, we would like to thank you again for your encouraging comments and positive evaluation of our work. Your constructive feedback has been invaluable in strengthening this paper.

---

> > > > > > > ### Author Response · Authors · 2025-08-07
> > > > > > > **Follow-up: Confirmation of Concerns Resolution**
> > > > > > >
> > > > > > > Thank you for your valuable feedback and positive discussion. Please let us know if you have any remaining concerns about our paper. We want to ensure all your questions have been fully resolved.

---

> ### Author Response · Authors · 2025-08-09
> **Thank you again for your supportive comments and your kind indication to raise our score**
>
> Thank you again for your supportive comments during the discussion phase and your kind indication to raise our score—we truly appreciate your recognition of our work.
>
> As the scoring system remains open for a short while longer, the updated score might not have been reflected in the system yet. If this was intentional, we completely understand, but we just wanted to check in case it was an oversight. Once again, we truly appreciate your valuable comments and your willingness to support our paper.
>
> Please feel free to disregard this note if you've already processed the update. We’re grateful for your time and consideration throughout this review process.

---

### Note · Authors · 2025-08-12

We thank the reviewers, ACs, and SACs for their valuable work. We are encouraged that reviewers find our paper has strong motivation toward on-device LLM-based ObjNav (Reviewer gaGt, LqYc, hd8c), our paper proposes valuable technical contributions (Reviewer QLLB, LqYc, hd8c), and our methods show good performance and overcome existing bottlenecks (Reviewer gaGt, QLLB, LqYc, hd8c). We are glad that our rebuttal clarifies the reviewers' concerns, and Reviewer gaGt and QLLB decided to improve their scores.

**Significance of this paper:** We propose EfficientNav to enable on-device efficient LLM-based zero-shot ObjNav. We propose discrete memory caching, which prevents saving the KV cache of the whole map description and meets the memory constraints. We propose attention-based memory clustering to reduce the impact of ignoring cross-attention between groups. We propose semantics-aware memory retrieval, which efficiently selects the relevant groups and prunes redundant map information.

**Summary of discussion:** We would like to emphasize following aspects:

1) Difference from sparse attention approaches: Our discrete memory caching can decouple the group order and KV cache computation of each group, thus enabling KV reuse with different group retrieval. However, for general sparse attention methods, their purpose is to minimize the difference between sparse attention and full attention and accelerate the attention calculation. But without memory retrieval, even full attention shows a low success rate. And sparse attention methods need to recompute the KV cache of the navigation map description because of map changes, thus showing longer real-time latency.

2) Robustness of EfficientNav: In this paper, we mainly focus on optimization of the planning process. And we also find our method shows good robustness with different detection models, different semantic matching models, and different numbers of layers used for attention-based clustering.

3) Deployment method: We deployed our model on the computation platform (Jetson Orin) that is frequently used by real-world robots, as our method mainly solves the problem of high computation cost in planning by optimizing KV cache management. And we would like to clarify that our simulation-based evaluation is also a widely used practice in ObjNav research, as seen in many prior works. Our experimental setup follows the same standard, ensuring fair comparison with existing methods while maintaining reproducibility.

---

### Decision · Program_Chairs · 2025-09-17

**Decision:**

Accept (poster)

**Comment:**

This paper proposes EfficientNav for on-device efficient LMM zero-shot ObjNav. Main contributions of the work are (1) discrete memory cashing to avoid recomputing KV cash (2) attention-based memory clustering and (3) semantic-aware memory retrieval to select relevant object groups based on navigation goal. Experiments show 11.1% improvement in success rate over GPT-4, while having 6.7x latency reduction. To further strengthen the paper authors should address all feedback of the reviewers, specifically improve overall writing/clarity, including more details on the overall architecture and model so it can be easy for reproduction; comparisons against baselines; practical deployment and evaluation in real world;